# Conformation of Pullulan in Aqueous Solution Studied by Small-Angle X-ray Scattering

**DOI:** 10.3390/polym12061266

**Published:** 2020-06-01

**Authors:** Jia Yang, Takahiro Sato

**Affiliations:** Department of Macromolecular Science, Osaka University, Toyonaka Osaka 560-0043, Japan; yangj17@chem.sci.osaka-u.ac.jp

**Keywords:** pullulan, amylose, small-angle X-ray scattering, scattering function, radius of gyration, wormlike chain

## Abstract

Small-angle X-ray scattering functions were measured for six pullulan samples with molecular weights ranging from 2.3 × 10^4^ to 7.4 × 10^5^ in 0.05 M aqueous NaCl at 25 °C and fitted by the perturbed wormlike chain model, comprising touched-bead sub-bodies, to obtain wormlike chain parameters. The parameter values determined were consistent with those determined from previously reported dilute solution properties of aqueous pullulan. Because radii of gyration of not only pullulan polymers, but also pullulan oligomers were consistently explained by the touched-bead wormlike chain model perturbed by the excluded volume effect, the pullulan chain takes a local conformation considerably different from the amylose chain, although both polysaccharides are flexible polymers with an approximately same characteristic ratio.

## 1. Introduction

Pullulan is a non-ionic water-soluble polysaccharide obtained from the fermentation of black yeast. Due to its non-toxic, non-immunogenic, and also environmentally friendly nature, pullulan is explored for various industrial and biomedical applications [1,2,3]. For example, Akiyoshi et al. [4,5,6,7,8] have developed hydrophobically modified pullulan nanogels, spontaneously forming complexes with proteins and hydrophobic drugs to use as a protein nanocarrier in vivo. Furthermore, because pullulan has high solubility to water and can be easily fractionated to obtain samples with narrow molecular weight distributions, pullulan is used as the calibration standard in aqueous size exclusion chromatography.

The molecular conformation of the pullulan chain in solution is often compared with that of amylose. Both pullulan and amylose are flexible d-glucans, and glucose residues of amylose are linked only through *α*-1,4 glucosidic bonds, but pullulan consists of trisaccharide repeating units connected by two *α*-1,4 bonds and a single *α*-1,6 bond. Due to the homogeneous glucosidic linkage, amylose is known to take a locally helical conformation, [9,10,11,12,13,14], but pullulan does not have such a helical nature, probably because of heterogeneity with respect to the glucosidic linkage. The better solubility of pullulan to water may be due to this non-helical nature.

Although the local conformation of pullulan has been studied theoretically [15,16,17], the locally helical nature of the pullulan conformation in solution has been little investigated experimentally so far. In the present study, we carried out small-angle X-ray scattering (SAXS) measurements on pullulan in aqueous solution (0.05 M aqueous NaCl) to reveal its local conformation.

There are a few reports of SAXS studies on pullulan in water. Muroga et al. [18] investigated the chain stiffness of pullulan in water by SAXS, from the crossover scattering wavenumber between the Debye and rod regions, but they did not analyze their SAXS profiles over the entire scattering wavenumber region examined in detail. Liu et al. [16] measured SAXS profiles for pullulan oligomers containing three, six, nine, and 12 glucose residues in water, and compared their results with the rotational isomeric state model. The present study connects their SAXS results for pullulan oligomers with those for pullulan polymers to reveal the non-helical nature of the pullulan chain in solution.

## 2. Materials and Methods

### 2.1. Sample and Solutions

Commercialized standard pullulan samples (Shodex Co., Tokyo, Japan), were used in this study. Their weight molecular weights *M*_w_ and molecular weight distributions *M*_w_*/M*_n_ reported by the sample supplier are listed in Table 1. Each pullulan sample was dissolved in 0.05 M aqueous sodium chloride (NaCl). NaCl was recrystallized from water and the solvent water was purified by a Millipore Milli-Q system.

### 2.2. SAXS Measurements

SAXS measurements were conducted on pullulan solutions at the BL40B2 beamline of SPring-8 (JASRI, Hyogo, Japan). Polymer mass concentrations *c* of the test solutions are listed in Table 1. The wavelength of the X-ray, the camera length and the accumulation time were set to be 0.1 nm, 4 m, and 180 s, respectively. A capillary made of quartz (2.0 mm inner diameter) that contained test solutions was set in a heating block at 25 °C, and the intensity of the scattered X-ray was measured using a PILATUS2M instrument (DECTRIS, Baden, Switzerland) and circularly averaged.

The excess Rayleigh ratio *R_θ,_*_X_ at the scattering angle *θ* and the optical constant *K*_e_ of SAXS were calculated by
(1)$$R_{\theta ,X}=F\left(\frac{I_{\theta ,\mathrm{soln}}}{I_{\mathrm{mon},\mathrm{soln}}}-\frac{I_{\theta ,\mathrm{solv}}}{I_{\mathrm{mon},\mathrm{solv}}}\right),K_{e}=N_{A}a_{e}{}^{2}\gamma^{2}$$
where *F* is the instrument constant, *I_θ_*_,soln_ (*I_θ_*_,solv_) and *I*_mon,soln_ (*I*_mon,solv_) are the scattering intensity at the scattering angle *θ* and the monitor value of the incident SAXS intensity, respectively, of the solution (of the solvent), *N*_A_ is the Avogadro constant, *a*_e_ is the classical radius of electron (=2.82 × 10^−13^ cm), and *γ* is the SAXS contrast factor of the polymer. The constant *F* was determined so as for the second virial coefficient *A*_2_ to be consistent with literature values (see below).

Pullulan can be used as the standard sample to calibrate the absolute intensity of SAXS from aqueous solutions. As explained in Appendix A, the contrast factor of pullulan in aqueous solution is larger than that of polyethylene glycol in the same solvent, so that pullulan is more suitable as the standard sample for SAXS from aqueous solutions.

### 2.3. Analysis of the SAXS Scattering Function [19]

Yoshizaki and Yamakawa [20,21] calculated the particle scattering function *P*(*k*) for the unperturbed wormlike chain, and proposed an empirical function to represent the calculation results as a function of the magnitude of the scattering vector *k* as well as the persistence length *q* and the Kuhn statistical number *N* (or the contour length *L* = 2*qN*) of the wormlike chain, given by
(2)$$P(k)=\left[\left(1-e^{-\xi^{-5}}\right)P_{{(C}^{*})}(k;{\langle S^{2}\rangle }_{0}^{1/2})+e^{-\xi^{-5}}P_{\left(R\right)}(k;L)\right]\Gamma (k;N,\xi )F_{0b}{}^{2}(kd_{b})$$
where 〈*S*^2^〉_0_^1/2^ is the radius of gyration of the unperturbed chain without the excluded volume effect, calculated by
(3)$${\langle S^{2}\rangle }_{0}=\frac{1}{3}qL-q^{2}+\frac{2q^{3}}{L}\left[1-\frac{q}{L}\left(1-e^{-L/q}\right)\right]$$
where *ξ* is defined by
(4)$$\xi \equiv \left(\frac{\pi }{2N}\right)\frac{{\langle S^{2}\rangle }_{0}}{{\left(2q\right)}^{2}}\left(2qk\right)$$
and *P*_(C_*_)_(*k*; 〈*S*^2^〉_0_^1/2^) and *P*_(R)_(*k*; *L*) are the particle scattering functions for the Gaussian coil (the Debye function) with the same 〈*S*^2^〉_0_^1/2^ as that of the wormlike chain, and the rod with the contour length *L*, respectively, given by
(5)$$P_{{(C}^{*})}(k;{\langle S^{2}\rangle }_{0}^{1/2})=\frac{2}{u^{2}}\left(e^{-u}+u-1\right)\;\left(u\equiv {\langle S^{2}\rangle }_{0}k^{2}\right),P_{\left(R\right)}(k;L)=\frac{2}{v^{2}}\left[v\mathrm{Si}(v)+\cos v-1\right]\;\left(v\equiv Lk\right)$$(Si(*v*): the sine integral). Numerical results for *P*(*k*) cannot be expressed by an average of *P*_(C_*_)_(*k*; 〈*S*^2^〉_0_^1/2^) and *P*_(R)_(*k*; *L*) in the bracket of Equation (2), so we need the correction factor Γ (*k*; *N*, *ξ*) in Equation (2) which is a function of *k*, *N*, and *ξ*, formulated by the original authors. The last function *F*_0b_(*kd*_b_) in Equation (2) is the correction factor of the finite thickness of the polymer chain, given by [22]
(6)$$F_{0}(kd_{b})=\frac{24}{{\left(kd_{b}\right)}^{3}}\left[\sin\left(\frac{1}{2}kd_{b}\right)-\left(\frac{1}{2}kd_{b}\right)\cos\left(\frac{1}{2}kd_{b}\right)\right]$$
where *d*_b_ is the bead diameter of the touched-bead wormlike chain model.

Pedersen and Schurtenberger [23] made off-lattice Monte Carlo simulations on the discrete wormlike chain with the excluded volume interaction to calculate the perturbed particle scattering function *P*(*k*). Calculated results for the perturbed *P*(*k*) are expressed in a similar form to Equation (2) for the unperturbed one. While the particle scattering function of the rod *P*_(R)_(*k*; *L*) is not affected by the excluded volume interaction, *P*_(C_*_)_(*k*; 〈*S*^2^〉^1/2^) is perturbed by the interaction, where 〈*S*^2^〉^1/2^ is the radius of gyration of the perturbed chain. Furthermore, the parameter *ξ* is replaced by
(7)$$\xi^{\prime }\equiv {\left(\frac{\pi }{1.103N}\right)}^{3/2}{\left(\frac{{\langle S^{2}\rangle }^{1/2}}{2q}\right)}^{2.564}\left(2qk\right)$$
and numerical coefficients in the correction function Γ (*k*; *N*, *ξ*) are dependent on the perturbation. The perturbed particle scattering function is given by
(8)$$P(k)=\left[\left(1-e^{-{\xi^{\prime }}^{-5}}\right){P^{\prime }}_{{(C}^{*})}(k;{\langle S^{2}\rangle }^{1/2})+e^{-{\xi^{\prime }}^{-5}}P_{\left(R\right)}(k;L)\right]\Gamma^{\prime }(k;N,\xi^{\prime })F_{0b}{}^{2}(kd_{b})$$
with the perturbed functions *P′*_(C_*_)_(*k*; 〈*S*^2^〉^1/2^) and Γ′(*k*; *N*, *ξ*) given by the original authors. The perturbed 〈*S*^2^〉^1/2^ can be calculated by the two parameters theory from
(9)$${\langle S^{2}\rangle }^{1/2}=\alpha_{S}(\tilde{z}){\langle S^{2}\rangle }_{0}^{1/2}$$
using the expansion factor *α*_S_ ($\tilde{z}$) of a known function of the scaled excluded-volume parameter $\tilde{z}$ defined by
(10)$$\tilde{z}=\frac{3}{4}K(N)\cdot {\left(\frac{3}{2\pi }\right)}^{3/2}BN^{1/2}$$Here, *K*(*N*) is the scale factor relating to the chain stiffness, and *B* expresses the excluded-volume strength. Functional forms of *α*_S_ ($\tilde{z}$) and *K*(*N*) are given by ref. [21,24].

At a finite polymer concentration, the scattering function is affected by the inter-molecular interference, and it may be written in the form [25].
(11)$$\frac{R_{\theta ,X}}{K_{e}c}=\frac{M_{w}P(k)}{1+2A_{2,\mathrm{app}}M_{w}P(k)c}$$
where *M*_w_ and *A*_2,app_ are the weight average molecular weight and the apparent second virial coefficient (including higher virial terms) for the polymer, and *P*(*k*) is calculated by Equation (2) or (8). Considering up to the third virial term, we can relate *A*_2,app_ to the true second virial coefficient *A*_2_ by
(12)$$A_{2}=\frac{\sqrt{1+6gA_{2,\mathrm{app}}M_{w}c}-1}{3gM_{w}c}$$
where the coefficient *g* (the reduced third virial coefficient) may be approximated to 1/3 for flexible polymers in good solvents [26].

Using the molar mass per unit contour length *M*_L_, we can calculate *L* from *M*_w_ by *L* = *M*_w_*/M*_L_. Thus, unknown parameters to calculate the SAXS scattering function *R_θ_*_,X_*/K*_e_*c* from Equations (8) and (11) are *M*_L_, *q*, *B*, *d*, and *A*_2,app_. Among them, *d* and *A*_2,app_ affect the scattering function only in high *k* and low *k* regions, respectively.

## 3. Results

### 3.1. Kratky Plot

Figure 1 shows the Kratky plots of six pullulan samples in 0.05 M aqueous NaCl at 25 °C. All the plots increase with *k* sharply in a low *k* region, and then gradually in higher *k* region. The Debye function P_(C_*_)_(*k*; 〈S^2^〉_0_^1/2^) for the Gaussian chain (given by Equation (5)) predicts that the Kratky plot becomes horizontal in high *k* region), but the plots in Figure 1 keep increasing with *k* even at high *k*. This is due to the non-Gaussian nature of the pullulan chains. On the other hand, the particle scattering function P_(R)_(*k*; *L*) for the rod (given by Equation (5)) indicates that the Kratky plot becomes proportional to *k* at high *k*. All the Kratky plots in Figure 1 do not possess such a k region within the *k* range investigated, probably because of the flexible nature of the global pullulan chain conformation.

Now, the Kratky plots in Figure 1 compare with Equation (11) along with Equation (8). The equations contain adjustable parameters *M*_L_, *q*, *B*, *d*, and *A*_2,app_. Because it was difficult to determine all the five adjustable parameters uniquly by the fitting, we assumed *M*_L_ to be 470 nm^−1^, which is the average of *M*_L_ determined from dilute solution properties reported previously (cf. Table 3). As mentioned above, *d* and *A*_2,app_ affect the scattering function only in high *k* and low *k* regions, respectively, while *q* and *B* alter the scattering function over entire *k* region. Thus, a trial-and error method was used to find *q* and *B* values which lead to the closest agreement between experiment and theory first, and then values of d and *A*_2,app_ were adjusted to obtain the best fit in high *k* and low *k* regions, respectively.

Red solid curves in Figure 1 show the best fits by Equations (8) and (11) for the perturbed wormlike chain. Parameter values used for the fittings are listed in Table 2. All the fittings are satisfactory over the entire *k* region examined. Black solid curves for samples P20 and P800 in the same figure are the scattering functions for the unperturbed wormlike chain calculated *P*(*k*) by Equation (2) instead of Equation (8) using parameter values listed in Table 2 (*B* = 0). Especially for the high molecular weight sample P800, the black solid curve remarkably deviates from the red solid curve for the same sample, demonstrating the importance of the excluded volume effect on the scattering function. Deviations of the dotted curves from the black solid curves for the same samples indicate the importance of the local chain stiffness effect on the scattering function in a high *k* region.

### 3.2. Radius of Gyration and Second Virial Coefficient

The perturbed radius of gyration 〈*S*^2^〉^1/2^ calculated by Equation (9) with the wormlike chain parameters determined are listed in the last line of Table 2. Those results are plotted against *M*_w_ double logarithmically in Figure 2a by unfilled red circles. On the other hand, unfilled black circles in the same figure represent Liu et al.’s 〈*S*^2^〉^1/2^ data of pullulan oligomers containing three, six, nine, and 12 glucose residues in 25 °C water obtained by SAXS [16]. The red and black circles are smoothly connected by the solid curve in the figure, which indicates theoretical values calculated by
(13)$$\langle S^{2}\rangle =\alpha_{S}{}^{2}(\tilde{z}){\langle S^{2}\rangle }_{0}+\frac{3}{20}d_{b}{}^{2}$$
with wormlike chain parameter values averaged over the six samples in Table 2 and the chain thickness (the bead diameter) *d*_b_ = 1.0 nm. Equation (13) is obtained from Equation (9) by the correction of the chain thickness using the touched-bead wormlike chain model [21]. This value of *d* is close to the results of *d* determined for SAXS scattering functions of high molecular weight pullulan samples (see the fifth line of Table 2) by using Equation (8), demonstrating that the chain thickness correction is successfully made by Equation (6) over the molecular weight range from oligomers to high polymers for pullulan.

Figure 2b compares the second virial coefficients *A*_2_ obtained in this study with those reported previously [27,28,29,30,31]. Red filled and unfilled circles indicate results of the apparent second virial coefficient *A*_2,app_ and the true second virial coefficient *A*_2_ corrected for the third virial term (cf. the sixth line of Table 2), respectively, in 0.05 M aqueous NaCl, and other symbols previously reported *A*_2_ obtained for pullulan in 25 °C water by light scattering and sedimentation equilibrium. Because pullulan is a non-ionic polysaccharide and also the NaCl concentration in this study is dilute enough, the difference in the solvent is expected not to affect *A*_2_. Although previous data are more or less scattered, our *A*_2_ results are consistent with the previous data.

### 3.3. Wormlike Chain Parameters Determined by Other Methods

Dilute solution properties, the radius of gyration 〈*S*^2^〉^1/2^, intrinsic viscosity [*η*], and the hydrodynamic radius *R*_H_, were studied for pullulan in water (25 °C) by several authors [27,28,29,30,31]. To compare those previous results with the present SAXS results, we analyze the previous data by the wormlike or touched-bead wormlike chain model [19,21]. Figure 3 shows fitting results for the previous data of [*η*], 〈*S*^2^〉^1/2^, and *ρ* = 〈*S*^2^〉^1/2^/*R*_H_. The wormlike chain parameters determined by the fittings are listed in Table 3. Solid curves for [*η*], 〈*S*^2^〉^1/2^, and *ρ* represent theoretical values for the perturbed (touched-bead) wormlike chain model, satisfactorily reproducing the experimental results, although Buliga and Brant’s 〈*S*^2^〉^1/2^ data points (red circles) [29] slightly deviate upward from the theoretical curve due to their pullulan samples with relatively broad molecular weight distributions. On the other hand, dotted curves for [*η*] and 〈*S*^2^〉^1/2^ are theoretical ones for unperturbed (touched-bead) wormlike chain model with the same (touched-bead) wormlike chain parameters except for *B* = 0, demonstrating that the excluded volume effect is important for pullulan in water at *M*_w_ higher than ca. 10^5^.

The solid curve in Figure 2b indicates theoretical values for *A*_2_ calculated by the Yamakawa theory [32] for the wormlike chain model with parameters, *M*_L_ = 460 nm^−1^, *q* = 1.8 nm, and *B* = 0.4 nm, without considering the chain end effect, which is important only in a low *M*_w_ region. The theoretical curve almost fits the present and previous data except for Kato et al.’s, [27,28] which slightly but systematically deviate downward from the curve.

Table 3 summarizes the wormlike chain parameters obtained from the present and previous data of dilute solution properties. All the parameter values determined in the present study are in good agreements with those from previous <*S*^2^>^1/2^ data. Agreements in *q*, *B*, and *d*_b_ of the present work with those determined from the hydrodynamic quantities are less satisfactory. Especially, the *d*_b_ values from [*η*] and *R*_H_ are considerably smaller than those from SAXS. While *d*_b_ in [*η*] and *R*_H_ is originated from the hydrodynamic boundary condition on the bead surface, *d*_b_ in SAXS is related to the radial distribution of the electron density around the bead. Both *d*_b_ should reflect the chain thickness, but are not necessarily guaranteed to agree quantitatively.

## 4. Discussion

Pullulan can be regarded as a periodic copolymer, where main-chain d-glucose residues are linked through two *α*-1,4-and a single *α*-1,6-glucosidic bonds. On the other hand, amylose is an *α*-1,4-d-glucan, where all d-glucoses are polymerized by *α*-1,4-glucosidic linkages. Thus, it may be intriguing to compare between conformations of pullulan and amylose.

Dilute solution properties of amylose in dimethylsulfoxide (DMSO) were analyzed by using the helical wormlike chain model by Nakanishi et al. [13] because of its main-chain helical nature. The helical wormlike chain is characterized in terms of *M*_L_, the stiffness parameter *λ*^−1^, intrinsic curvature *κ*_0_, and intrinsic torsion *τ*_0_ [21]. While the conformation at the energy minimum is a straight rod for the wormlike chain, that for the helical wormlike chain is an intrinsic helix determined by *κ*_0_ and *τ*_0_. Thus, the chain contour of the helical wormlike chain is not necessarily identical with that of the wormlike chain, and moreover *λ*^−1^ cannot be directly compared with *q* of the wormlike chain at *κ*_0_ ≠ 0, though it is equal to 2*q* in the limit of *κ*_0_ = 0. Nakanishi et al. determined the helical wormlike chain parameters using intrinsic viscosity data in a low molecular weight region, where the excluded volume effect is not important. Parameter values determined are *M*_L_ = 500 nm^−1^, *λ*^−1^ = 4 nm, *λ*^−1^*κ*_0_ = 3.5, and *λ*^−1^*τ*_0_ = 4.0. The non-zero value of *κ*_0_ indicates the helix nature of amylose, and the pitch and diameter of the amylose intrinsic helix are 3.6 nm and 0.50 nm, respectively, indicating an extended helical conformation at the energy minimum. However, the *λ*^−1^ value is so small that the intrinsic helical conformation is considerably disorganized.

As shown in Figure 2a, <*S*^2^>^1/2^ data for pullulan oligomers and polymers are perfectly explained in terms of the wormlike chain including the chain thickness correction. Thus, we can say that the pullulan chain has no helical nature, probably due to mingling *α*-1,6-glucosidic linkage. Using the molecular weight of the glucose residue *M*_0_ = 162, the contour length per glucose residue *h* = *M*_0_*/M*_L_ is 0.35 nm for pullulan and 0.32 nm for amylose. The small difference in *h* may reflect differences between *α*-1,6- and *α*-1,4-glucosidic linkages as well as in the helix nature.

The characteristic ratio *C_n_* of pullulan and amylose is calculated by [33]
(14)$$C_{n}=\frac{6M_{0}}{b_{u}{}^{2}}\frac{{\langle S^{2}\rangle }_{0}}{M}$$
where *M*_0_ is the molecular weight of the glucose residue (=162), *M* and *n* (=*M/M*_0_) are the molecular weight and number of glucose residues of the polysaccharide, and *b*_u_ is the virtual bond length (i.e., the distance between oxygen atoms of the neighboring glucosidic bonds; for pullulan, the root-mean-square of three sequential virtual bond lengths in the trimer unit). For pullulan, *b*_u_ is 0.479 nm [29], and for amylose 0.425 nm [10]. Table 4 compares characteristic ratios *C*_∞_ at infinite molecular weight for pullulan and amylose. The chain conformation of amylose was characterized not only in DMSO but also in 0.33 M aqueous KCl (a theta solvent) [34,35] and in water [36]. All the results of *C*_∞_ for pullulan and amylose are within 4.9 ± 0.4, demonstrating that both polysaccharides take random coil conformations. Due to the longer *b*_u_, the dimension of pullulan is slightly larger than amylose with the same molecular weight.

Table 4 also contains theoretical results of *C*_∞_ for pullulan and amylose. The old result of Rao et al. [37] is slightly larger than, but other theoretical results [12,15] are in good agreements with the experimental results. Brant et al. [15,16] demonstrated that theoretical results of *C_n_* for pullulan oligomers also agree with experimental results.

As shown in Figure 2a, 〈*S*^2^〉^1/2^ data for pullulan oligomers and polymers are explained in terms of the wormlike chain including the chain thickness correction. This figure forms a striking contrast to the degree of polymerization *n* dependence of *C_n_* for amylose given by Fujii et al. [11] using Jordan et al.’s rotational isomeric state model calculation [10], where *C_n_* exhibits oscillations, indicating the local helical nature, in the range of *M* < 5000 (cf. Figure 9 of ref [11]). Thus, we can say that the local conformation of the pullulan chain is definitely different from that of the amylose chain, probably due to mingling *α*-1,6-glucosidic linkage, although *C*_∞_ of the two polysaccharides in the high *M* limit are indistinguishable. Using the molecular weight of the glucose residue *M*_0_ = 162, the contour length per glucose residue *h* = *M*_0_*/M*_L_ is 0.35 nm for pullulan and 0.32 nm for amylose. The difference in the local conformation between pullulan and amylose may be compensated by the difference in *h* to provide equal *C*_∞_.

## 5. Conclusions

SAXS profiles for dilute aqueous solutions of pullulan were fitted by the perturbed wormlike chain model considering the chain thickness effect to determine the wormlike chain parameters. The parameters were consistent with those determined by other dilute solution properties of pullulan in water, previously reported. Radii of gyration determined by SAXS for pullulan polymers and oligomers were explained by the touched-bead wormlike chain model with a 1 nm bead diameter over the entire molecular weight range examined. Since the virtual bond length of pullulan is ca. 0.5 nm, the disaccharide unit of pullulan is viewed as the scattering-unit sub-body. Differing from the amylose chain, we cannot expect the locally helical nature disappears for the pullulan chain within the disaccharide-unit resolution, probably by mingling *α*-1,6-glucosidic linkage.

## Figures and Tables

**Figure 1 polymers-12-01266-f001:**
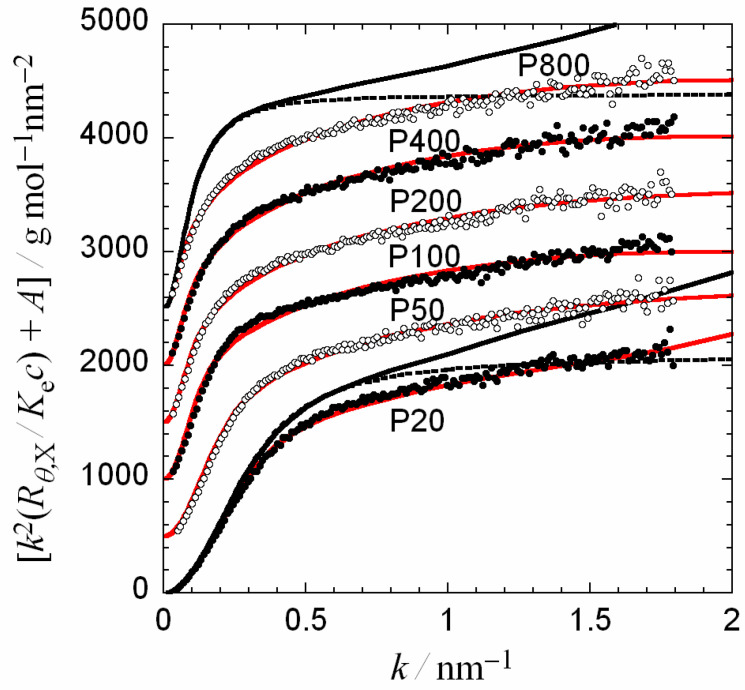
Kratky plots of pullulan samples in 0.05 M aqueous NaCl at 25 °C. The scattering functions except for sample P20 shift vertically by the shift constant *A* = 500, 1000, 1500, 2000, and 2500 for samples P50, P100, P200, P400, and P800, respectively. Red solid curves are theoretical scattering functions calculated by Equations (8) and (11) for the perturbed wormlike chain with parameter values listed in Table 2, while black solid and dotted curves for samples P20 and P800 indicate theoretical values calculated by Equations (2) and (11) for the unperturbed wormlike chain, and using *P*(*k*) = *P*_(C_*_)_(*k*; <*S*^2^>_0_^1/2^) in Equation (2) for the unperturbed thin Gaussian chain, respectively.

**Figure 2 polymers-12-01266-f002:**
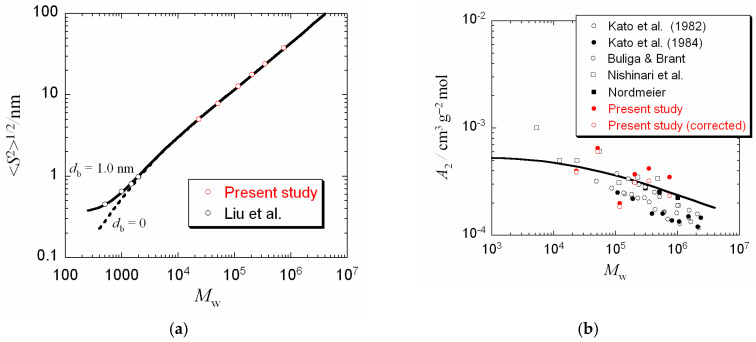
Molecular weight dependences of the perturbed radius of gyration 〈*S*^2^〉^1/2^ (**a**) and the second virial coefficient *A*_2_ (**b**) of pullulan in aqueous solution at 25 °C. Solid and dashed curves indicate theoretical values for the wormlike chain model (see the text).

**Figure 3 polymers-12-01266-f003:**
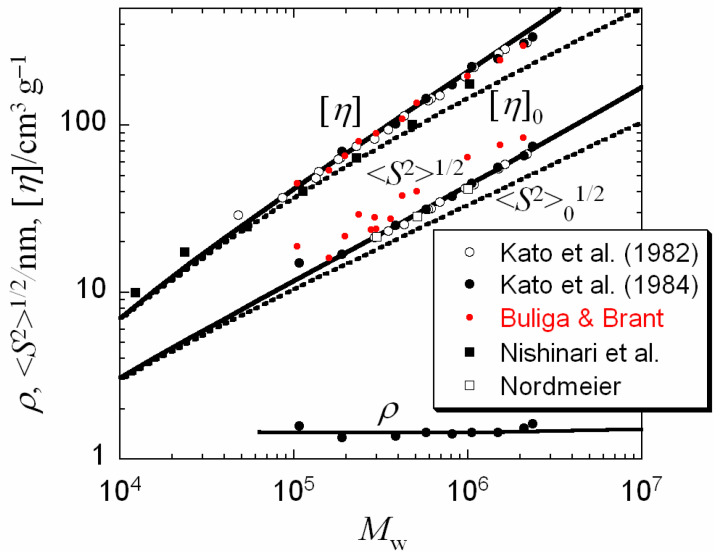
Molecular weight dependences of [*η*], 〈*S*^2^〉^1/2^, and *ρ* (= 〈*S*^2^〉^1/2^/*R*_H_) for pullulan in water at 25 °C obtained by Kato et al. [27,28], Buliga and Brant [29], Nishinari et al. [30], and Nordmeier [31], being compared with the theory of the perturbed (solid curves) and unperturbed (dotted curves) wormlike chain model.

**Table 1 polymers-12-01266-t001:** Molecular weights and molecular weight distributions of pullulan samples used in this study.

Sample	P20	P50	P100	P200	P400	P800
*M*_w_/10^4^	2.30	5.06	11.6	20.2	34.3	73.6
*M*_w_/*M*_n_	1.08	1.09	1.11	1.31	1.30	1.23
*c*/10^−3^ g cm^−3^	6.84	5.11	7.61	6.16	5.48	5.79

**Table 2 polymers-12-01266-t002:** Wormlike chain parameters of 6 pullulan samples in 0.05 M aqueous NaCl determined from SAXS scattering functions.

Sample	P20	P50	P100	P200	P400	P800
*M*_L_/nm^−1^	470 ^a^	470 ^a^	470 ^a^	470 ^a^	470 ^a^	470 ^a^
*q*/nm	1.5	1.5	1.6	1.6_5_	1.6_5_	1.5
*B*/nm	0.35	0.30	0.30	0.38	0.35	0.52
*d*/nm	0.77	0.77	0.89	0.77	0.89	0.84
*A*_2,app_ (*A*_2_ ^c^) ^b^	4.0 (3.9)	6.5 (6.0)	2.0 (1.8_5_)	3.7 (3.1)	4.2 (3.2)	3.5 (2.3)
<*S*^2^>^1/2^ (<*S*^2^>_0_^1/2^)/nm ^d^	4.9 (4.7)	7.6_5_ (7.1)	12._5_ (11)	18 (15)	24 (20)	37 (28)

^a^ Assumed value. ^b^ In units of 10^−4^ cm^3^ g^−2^ mol. ^c^ Calculated by Equation (12). ^d^ Calculated by Equations (3) and (9).

**Table 3 polymers-12-01266-t003:** Wormlike chain parameters of pullulan in water and 0.05 M aqueous NaCl at 25 °C, determined by different methods.

Method	SAXS ^a^ (Present Study)	〈*S*^2^〉^1/2 b^	[*η*] ^b^	*R* _H_ ^b^	*A* _2_ ^b^
*M*_L_/nm^−1^	470 ^c^	480	470	450	460
*q*/nm	1.6	1.6	1.8	2.0	1.8
*B*/nm	0.37	0.45	0.25	0.5	0.4
*d*_b_/nm	0.83	1.0 ^d^	0.45	0.50	-

^a^ In 0.05 M aqueous NaCl at 25 °C. ^b^ In 25 °C water. ^c^ Assumed value. ^d^ Determined from oligomer sample data [16].

**Table 4 polymers-12-01266-t004:** Characteristic ratios of pullulan and amylose.

Polymer	Pullulan	Amylose		
solvent	in water and 0.05 M NaCl	in DMSO	in 0.33 M KCl	in water
*C*_∞_ (experiment)	4.7	4.5 [13]	5.3 [34,35]	4.5 [36]
*C*_∞_ (theory)	4.7 [15,16]	6.9 [37], 4.5 [15], 5.0 [12]		

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
