# Peer review of "Conformation of Pullulan in Aqueous Solution Studied by Small-Angle X-ray Scattering"

_polymers, 2020, doi:10.3390/polym12061266_

Round 1
Reviewer 1 Report
The authors use Small angle X-Ray Scattering to investigate the perturbed radius of gyration and the second virial coefficient of pullulan in aqueous solution. The samples were kept at room temperature. The authors applied a parametric perturbed wormlike chain model. From the "excellent" fits the authors conclude that pullulan has no helical nature.
The work is well written and should be published.
I have some remarks and questions left.
The fits in the low Q regime do not fit the data the excellent. Please comment. Figure 1 P100 does not fit that well.
What were a parametric model for a helical structure?
What is the relation between the second osmotic coefficient and the radius of gyration?
Can the authors complement their findings with absorption data.
Author Response
- The fits in the low Q regime do not fit the data the excellent. Please comment. Figure 1 P100 does not fit that well.
The disagreement between the experiment and theory in Figure 1 for sample P100 is so small that we do not think to need some comment on the disagreement.
- What were a parametric model for a helical structure?
The helical wormlike chain can represent the helical nature of linear polymer chains in terms of the two parameters κ0 and τ0, from which one can calculate the characteristic helical pitch and diameter; please refer Yamakawa’s textbook [ref. 21]. At κ0 = 0 (the wormlike chain limit), the characteristic helical pitch tends to infinity, i.e., no helical nature. All the dilute solution properties for pullulan investigated in this paper can be explained by the wormlike chain limit with κ0 = 0, so that we conclude that pullulan chain has no helical nature.
- What is the relation between the second osmotic coefficient and the radius of gyration? Can the authors complement their findings with absorption data.
Both second virial coefficient and radius of gyration for pullulan can be fitted by the perturbed wormlike chain model, as shown by solid curves in Figure 2, demonstrating that the relation between the two quantities can be explained consistently by this model. Because pullulan is UV-vis inactive, we cannot discuss our findings with absorption data.
Reviewer 2 Report
This is a good paper that adds significantly to what is known about the aqueous solution behavior of the carbohydrate polymer in question, pullulan. The authors have taken adequately into account the relevant prior work on this system. Their experimental studies supplement the existing published work, and their interpretation of the new studies is effectively discussed in the context of the prior work. The authors have chosen to interpret the collected body of experimental studies (the prior work plus their own) in terms of a perturbed touched-bead wormlike chain model, which they convincingly demonstrate correlates well with the experimental data using a physically reasonable choice of the several parameters of the model. I noticed with reference to lines 134 and 137 that the authors appear to have referred in error to equation 4.
Author Response
- I noticed with reference to lines 134 and 137 that the authors appear to have referred in error to equation 4.
Thank you for your comments. We have corrected with reference the line 134 and 137 from equation 4 to equation 5 in the revised manuscript.